# H^+^-Translocating Membrane-Bound Pyrophosphatase from *Rhodospirillum rubrum* Fuels *Escherichia coli* Cells via an Alternative Pathway for Energy Generation

**DOI:** 10.3390/microorganisms11020294

**Published:** 2023-01-23

**Authors:** Evgeniya A. Malykh, Liubov I. Golubeva, Ekaterina S. Kovaleva, Mikhail S. Shupletsov, Elena V. Rodina, Sergey V. Mashko, Nataliya V. Stoynova

**Affiliations:** 1Ajinomoto-Genetika Research Institute, 117545 Moscow, Russia; 2Computational Mathematics and Cybernetics Department, Lomonosov Moscow State University, 119991 Moscow, Russia; 3Chemistry Department, Lomonosov Moscow State University, 119991 Moscow, Russia; 4Biological Department, Lomonosov Moscow State University, 119991 Moscow, Russia

**Keywords:** inorganic pyrophosphatase, *Rhodospirillum rubrum*, ^13^C-MFA, H^+^-translocating pyrophosphatase, soluble pyrophosphatase, metabolic engineering, pyrophosphate hydrolysis, ATP, synthetic biology

## Abstract

Inorganic pyrophosphatases (PPases) catalyze an essential reaction, namely, the hydrolysis of PP_i_, which is formed in large quantities as a side product of numerous cellular reactions. In the majority of living species, PP_i_ hydrolysis is carried out by soluble cytoplasmic PPase (S-PPases) with the released energy dissipated in the form of heat. In *Rhodospirillum rubrum,* part of this energy can be conserved by proton-pumping pyrophosphatase (H^+^-PPase^Rru^) in the form of a proton electrochemical gradient for further ATP synthesis. Here, the codon-harmonized gene *hppa*^Rru^ encoding H^+^-PPase^Rru^ was expressed in the *Escherichia coli* chromosome. We demonstrate, for the first time, that H^+^-PPase^Rru^ complements the essential native S-PPase in *E. coli* cells. ^13^C-MFA confirmed that replacing native PPase to H^+^-PPase^Rru^ leads to the re-distribution of carbon fluxes; a statistically significant 36% decrease in tricarboxylic acid (TCA) cycle fluxes was found compared with wild-type *E*. *coli* MG1655. Such a flux re-distribution can indicate the presence of an additional method for energy generation (e.g., ATP), which can be useful for the microbiological production of a number of compounds, the biosynthesis of which requires the consumption of ATP.

## 1. Introduction

Inorganic pyrophosphatases (PPases), EC3.6.1.1, are essential enzymes that are ubiquitous in all living organisms. PPases catalyze an essential reaction, the hydrolysis of the inorganic pyrophosphate (PP_i_) into two molecules of inorganic orthophosphate (P_i_). In a living cell, PP_i_ is formed in large quantities as a side product of multiple biosynthetic reactions, such as the synthesis of cell polymers, coenzymes, proteins, nucleic acids, and amino acids. Thus, PPases play an important role in cellular metabolism, shifting the equilibrium of these important reactions [1], such that they become practically irreversible. 

Presently, PPases in a wide variety of organisms can be divided into two large different classes: soluble (S-PPases, soluble families I, II, and III) and membrane-bound (M-PPase) (Figure 1). Members of families I and III, with very rare exceptions, consist of single-domain subunits, whereas subunits of the first discovered family II PPases comprise two domains [2]. The family III S-PPases, which are in fact modified haloalkane dehalogenases, have not been extensively characterized and are present in only a few bacterial species.

Of special interest is the class of M-PPases (EC7.1.3.1, H^+^-exporting diphosphatase; EC7.1.3.2, Na^+^-exporting diphosphatase), which differs essentially from S-PPases in architecture and biological function. M-PPases belong to an evolutionarily ancient protein family [4,5,6] and act as primary ion pumps in plants, algae and some protozoans, bacteria, and archaea that producing proton and/or sodium gradients [7,8]. M-PPases are among the simplest membrane transporters and have no homologs among known ion pumps [9]. These unique enzymes couple the reversible reaction of phosphorolysis/synthesis of PP_i_ to H^+^ and/or Na^+^ pumping across cytoplasm. Notably, in all known cases, species possessing M-PPase also contain S-PPase, which can be explained by the need to adapt to different growth conditions. Baykov et al. noted that soluble PPase belonging to any of its three unrelated families may coexist with a membrane PPase(s), but family I PPase is the most frequent companion. Of family II PPases, only the regulatable subfamily, whose members contain nucleotide-binding cystathionine β-synthase domains, coexists with M-PPases [2]. The interest in bacterial M-PPases significantly increased when the first known gene coding for H^+^-PPase/H^+^-PP_i_ synthase from the photosynthetic bacterium *Rhodospirillum rubrum* was characterized [7]. To date, many different M-PPases from plants and bacteria have been described; some have been expressed in *Escherichia coli* [10,11] and *Saccharomyces cerevisiae* [12] cells, and their physiological roles have also been studied [13,14,15].

Significant progress has been made in recent years in understanding the structural and functional properties of M-PPases and the mechanism through which these PPases couple PP_i_ hydrolysis with cation transport [9,16,17,18,19].

Here, the novel function of *R. rubrum* H^+^-PPase in terms of its bioenergetic advantages is suggested. In the majority of living species, hydrolysis of PP_i_ is accomplished by an S-PPase that dissipates released energy as heat. H^+^-PPase from *R. rubrum* can conserve part of this energy in the form of the proton electrochemical gradient. Thus, we attempted to utilize the high-energy bonds of PP_i_, releasing mainly in cell polymers synthesis and other cellular processes, for H^+^-pumping and for subsequent ATP synthesis instead of allowing this energy to be dissipated.

ATP is of high importance in a wide range of biotechnological processes aimed at producing cell metabolites or polypeptides. The rapid development of metabolic engineering makes it possible to achieve these goals more rationally and efficiently [20,21]. It is already becoming apparent that the regeneration of cofactors used in the target product synthesis pathway is required, along with traditional approaches, for example, increasing the expression levels of the corresponding genes, the attenuation of undesirable pathways, etc. It is well known that cofactors such as ATP/ADP [22,23,24], NADH/NAD^+^ [25], and acetyl CoA (coenzyme A)/CoA [26], which are shared among metabolic pathways, play central roles in the distribution and rate of metabolic fluxes. ATP is involved in many metabolic pathways and is essential for the microbial production of a wide range of industrially valuable compounds. Thus, attempts to provide ATP supply could be a powerful tool to obtain a number of ATP-consuming cellular metabolites, including some amino acids [21].

In the present study, *E. coli* cells were engineered by the heterologous expression of *R. rubrum* H^+^-PPase (H^+^-PPase^Rru^). The in vivo functionality of H^+^-PPase^Rru^, as the only inorganic pyrophosphatase in *E. coli* cells was first confirmed in this work by replacing the native soluble cytoplasmic *E. coli* PPase, normally necessary for survival [27], with membrane-bound H^+^-PPase^Rru^.

The hypothesis of the bioenergetic benefits of H^+^-PPase expression in *E. coli* cells was confirmed by ^13^C-metabolic flux analysis (^13^C-MFA).

## 2. Materials and Methods

### 2.1. Strains, Plasmids, and Media

All of the bacterial strains and plasmids used in this study are listed in Table 1. The *E. coli* strains were cultivated on LB (lysogeny broth), SOB (super optimal broth), SOC (super optimal broth with catabolite repression), or M9 medium (minimal medium with the glucose (0.4%) as a carbon source) [28]. The following antibiotics, chloramphenicol (Cm, 20 mg/L), ampicillin (Ap, 100 mg/L), kanamycin (Km, 50 mg/L), and tetracycline (Tc, 20 mg/L), were added into the medium as needed.

### 2.2. DNA Manipulation

All of the techniques for the manipulation and isolation of nucleic acid, as well as genetic manipulation with *E. coli* cells, were carried out in accordance with the standard protocols [28]. The following commercially available reagents were used in this study: 1 kb DNA Ladder, Taq polymerase, T4 DNA ligase, High Fidelity PCR Enzyme Mix, and restriction enzymes (Thermo Scientific Inc., Waltham, MA, USA). Oligonucleotides were synthesized by Evrogen (Moscow, Russia). The sequences of the oligonucleotide primers used in this study are presented in Table 2.

### 2.3. Construction of the Plasmid pAH162-Tc^R^-2Ter-hppa^Rru^

The sequence of the structural part of the gene *hppa*^Rru^, which is native for *R*. *rubrum*, was codon-harmonized for expression in *E*. *coli,* chemically synthesized, and cloned on the commercially available vector pUC57 (GenScript^®^, Piscataway, NJ, USA). Codon harmonization, based on codon-usage databases for *E. coli* and *R. rubrum* (*Kazusa.or.jp* source), was performed; codons were altered to maintain the frequency of their usage in *E. coli,* similar to that in *R. rubrum*; the complete DNA sequence is presented in Appendix B. The *Pst*I restriction site having at the 5′-end the nucleotide sequence *ccaaatt* and the *Sac*I having at the 3′-end the nucleotide sequence *atcccaaatt* were added for further re-cloning the *hppa*^Rru^ gene into the integrative vector pAH162-Tc^R^-2Ter [31].

The nucleotide sequence of the chemically synthesized *hppa*^Rru^ gene was verified by sequence analysis using the oligonucleotide primers P1 and P2, and DNA from the pUC-57-*hppa*^Rru^ plasmid as a template. The recombinant plasmid pUC57-*hppa*^Rru^ was treated with *Pst*I and *Sac*I restriction enzymes to re-clone *hppa*^Rru^ into the integrative vector pAH162-Tc^R^-2Ter treated with the same restriction enzymes. The resulting recombinant plasmid pAH162-Tc^R^-2Ter-*hppa*^Rru^ was obtained after transformation of the CC118λ*pir*^+^ *E*. *coli* strain (ATCC BAA 2426 [30]) and purification from one of the independently grown Tc^R^ clones. After verification of the desired recombinant structure, the resulting plasmid, pAH162-Tc^R^-2Ter-*hppa*^Rru^, was used for ϕ80-mediated integration into the artificial ϕ80-*attB* site on the chromosome of *E*. *coli* MG1655 Δ(ϕ80-*attB*) *IS*5.8::ϕ80-*attB* strain constructed earlier [31], according to the Dual-In/Out strategy [31,33].

### 2.4. Construction of Strains

The introduction of genetic modifications into the *E. coli* chromosome were performed using the λRed Recombineering system according to the Datsenko and Wanner method [32]. For this, we used the plasmid pKD46, harboring the arabinose-inducible λRed genes. The ϕ80-dependent site-specific integration of DNA fragments into the *E. coli* chromosome was carried out according to the method of Haldimann and Wanner using the pAH123 helper-plasmid providing thermoinducible expression of the ϕ80-*int* gene [33]. The cassettes with the Cm^R^, Tc^R^, and Km^R^ markers flanked by hybrid λ*attL* and λ*attR* (*λattP/λattB*-sites) were specially designed and used as selectable markers for further introduction into *E. coli* strains by P1-general transduction [36]. To eliminate the antibiotic resistance (Cm^R^, Tc^R^ and Km^R^ markers) from the *E. coli* chromosome, the helper plasmid pMW-λInt/Xis, which provides the λXis/Int site-specific recombination, was used [31]. The details of the mutant strain construction are presented in Appendix C, Appendix D and Appendix E.

### 2.5. Isolation of E. coli Inverted Membrane Vesicles (IMV)

The IMV were isolated according to Belogurov [10] with modifications. Cells of *E. coli* cultured in 5 mL LB were grown during the night. Then, 1 mL of fresh cells was subcultured in 50 mL LB/LB + Cm 40 mg/L and grown in a flask until reaching the logarithmic phase (OD_600_ = 0.8). Then, the cells from 50 mL of culture were washed in 0.9% NaCl (+4 °C), and the washed cell pellets were frozen and stored at −70 °C. Next, the cells were resuspended in 25 mL of ice-cold buffer A (120 mM Tris-HCl, 40 µM EGTA (egtazic acid), 2 mM MgS0_4_, 10% glycerol, pH 7.5) and pelleted by centrifugation at 8000× *g* for 30 min at 4 °C. The cell pellet was resuspended in 25 mL of buffer A (120 mM Tris-HCl, 40 µM EGTA, 2 mM Mg^2+^, 10% glycerol, pH 7.5) and disrupted using a single passage through a French pressure cell with a big eyelet at 2000 psi. Unbroken cells and cell debris were removed by centrifugation at 8000 × *g* for 30 min at 4 °C. The supernatant was supplemented with DNase I (1 µL of 1000 U/25 mL), transferred to an ultracentrifuge bottle (2.5 mL of supernatant, analytical balanced) and centrifuged at 150,000× *g* for 3 h at 4 °C; the procedure of washing and ultracentrifugation was repeated twice. After ultracentrifugation, the purified IMVs were resuspended in buffer A (1 mL), then aliquoted (50 µL), frozen, and stored at −70 °C until use. Protein concentrations in IMV suspensions were measured using the Bradford assay [37]. IMV quantities were calculated in terms of the protein content.

### 2.6. PP_i_ Hydrolysis Measurement of R. rubrum H^+^-PPase^Rru^

Measurement of the H^+^-PPase^Rru^ activity of IMV was performed using a commercially available P_i_Per™ Phosphate Assay Kit (Molecular Probes, Eugene, OR, USA). The following protocol was used for the ultrasensitive assay that detects free inorganic phosphate in solution through the formation of the fluorescent product resorufin (absorption/emission maxima ∼563/587 nm) and can be adapted for monitoring the kinetics of P_i_-generating enzymes [38,39]. For the in vivo assay, 120 mM Tris-HCl buffer (pH 8.0), 2 mM MgCl_2_, 0.5 mM NaF, and IMV suspension were mixed in a UV-star microplate (Greiner Bio-One, Kremsmünster, Austria) with the components of the P_i_Per™ Phosphate Assay Kit. 0.1 mM NaPP_i_ (Sodium pyrophosphate tetrabasic decahydrate, Merck, Germany) was added to initiate the reaction. The kinetics of the H^+^-PPase^Rru^ activity was monitored for 30 min. The formation of phosphate is expressed in a.u. ((F530/590) (ex/em)/mkg of IMV).

### 2.7. PP_i_ Hydrolysis Measurement of E. coli PPase

Measurement of the *E. coli* PPase activity was performed according to Heinonen and Lahti with modifications [40]. Cells of *E. coli* cultured in 5 mL M9 medium supplemented with LB medium (1/10 LB/M9) were grown until reaching the logarithmic phase and were used for enzyme preparation. The cells were washed with physiological solution and frozen at −20 °C until use.

Cell pellets were suspended in 50 mM Tris-HCl buffer, pH 8.0, containing 4 mg/mL of MgSO_4_·7H_2_O and 7.5 mg/mL of KCl and disrupted by sonication for 2 × 30 s with cooling. The debris was removed by centrifugation at 8000× *g* for 20 min (4 °C). Protein concentration was determined using Bradford assay [37]. For the assay, 50 mM Tris-HCl buffer, pH 8.0 was mixed in a test tube with cell lysate. The reaction was started by adding of 20 µL of 50 mM PP_i_ (sodium pyrophosphate tetrabasic decahydrate, Merck, Germany), followed by incubation for 30 min. After incubation, the reaction was stopped by adding 20 µL of 1 M citric acid. A reaction mixture, where citric acid was added before PP_i_, was used as a blank.

P_i_ was determined as follows: into test tubes containing 0.125 mL each of sample (50–1500 nmol of P_i_), 1 mL of the AAM solution (acetone/5 N sulfuric acid/10 mM ammonium heptamolybdate 2/1/1) was added. The contents were mixed carefully using a vortex mixer. Then, 0.1 mL of 1 M citric acid was pipetted into each tube. After mixing again, absorbance of the yellow color was measured at 350 nm in UV-star microplate (Greiner Bio-One, Austria). A sample with no added cell was used as a blank. For calibration curves, samples with standard solutions of KH_2_PO_4_ were used. The specific activity was determined using the formula:(1)$$A\;\left(\frac{\mu \mathrm{mol}}{t\;\min*\mathrm{mg}\;\mathrm{of}\;\mathrm{protein}}\right)=\left(\frac{c\left(P_{i}\right)\;\mathrm{mmol}}{t\;\min*\mathrm{mg}\;\mathrm{of}\;\mathrm{protein}*1000}\right)$$
where t is the time of the reaction.

### 2.8. Southern Blotting Analysis

Southern hybridization was performed in accordance with the known method [28] using the following equipment: BrightStar™-Plus Positively Charged Nylon Membrane (Thermo Fisher Scientific, Waltham, MA, USA), VacuGene XL Vacuum Blotting System (GE Healthcare, Chicago, IL, USA), and a Hybridization Oven/Shaker (former Amersham Biosciences). DNA labeling with Biotin-11-dUTP (Thermo Fisher Scientific, Waltham, MA, USA) was performed in a standard 50 μL PCR reaction with the necessary pairs of primers and templates and 0.2 mM of biotin labeling mix and Taq DNA polymerase. The biotin labeling mix consisted of 2 mM dGTP, 2 mM dATP, 2 mM dCTP, 1.3 mM dTTP, and 0.7 mM of the Biotin-11-dUTP aqueous solution. Biotin chromogenic detection kits (Thermo Fisher Scientific, Waltham, MA, USA) were used to detect the DNA probes after Southern hybridization. The oligonucleotide primers P3 and P4 were used for the PCR amplification of the probes for the *ppa* gene (Table 2).

### 2.9. ^13^C-MFA

#### 2.9.1. Carbon Labeling Experiment

Carbon labeling experiment (CLE) was carried out in 150 mL microjars (ABLE Biott., Japan) containing 55 mL of synthetic medium of following composition: 1 × M9 salts (M9 minimal salts, Merck, Germany); 3 g/L glucose; 2 mM MgSO_4_; 0.8 mM CaCl_2_; 4.9 mM thiamine; 0.064 mM FeSO_4_·7H_2_O; 0.00613 mM CuSO_4_·5H_2_O; 0.00626 mM ZnSO_4_·7H_2_O; 0.00834 mM MnSO_4_·5H_2_O; 0.00869 mM CoCl_2_·6H_2_O; 0.545 mM Na_2_EDTA·2H_2_O. The 100% [1,2-^13^C] glucose (Cambridge Isotope Laboratories, Inc., USA) was used as a labeled substrate. The pH was maintained at 7.0 by NH_3_ gas. Mixing at 700 rpm and simultaneous atmospheric air flow (55 mL/min) were used for aeration. The seed cultures were prepared in several steps. First, cells grown overnight in 750 mL flasks with 30 mL of synthetic medium were inoculated into new flasks with fresh medium of the same composition and were cultivated with aeration until the mid-logarithmic growth phase (OD_595_ of 0.5–0.6). Then, these cultures were used to inoculate the microjars with an initial biomass concentration of 0.01 optical density (OD) at 595 nm (OD_595_). Cultivation at all steps was carried out at 37 °C. Samples were taken periodically to monitor the optical density (OD_595_) of cultures and the concentrations of glucose and acetate.

#### 2.9.2. Analysis of Substrate and Products

The OD_595_ was measured using a Biotek Synergy 2 microplate reader (Biotek Instruments, Winooski, VT, USA).

The glucose concentration was measured using Biosen C-Line Glucose Analyzer (EKF—Diagnostic GmbH, Barleben, Germany).

The ion-exchange chromatography combined with the pH-buffered electroconductivity method (Shimadzu HPLC Manual, Chiyoda City, Japan) was used to determine the acetate concentration.

#### 2.9.3. Metabolic Map

The “consensus” *E*. *coli* core metabolic model [41] was used with minor changes (see Appendix A). Briefly, the model included the main pathways of *E. coli* glucose metabolism: Embden−Meyerhof−Parnas (EMP), pentose−phosphate (PP), and Entner−Doudoroff (ED), the tricarboxylic acid (TCA) cycle, the glyoxylate shunt, anaplerotic carboxylation, and decarboxylation of malate and oxaloacetate. For the transketolase (EC 2.2.1.1, TK) and transaldolase (EC 2.2.1.2, TA) reactions, a ping-pong mechanism of their action was considered, and reactions were modeled as metabolite specific, reversible, C2 and C3 fragment producing, and consuming half-reactions of TK-C2 and TA-C3 [42].

The fructose-1,6-bisphosphatase and phosphoenolpyruvate synthetase reactions were added to the model, as cells grown on glucose minimal medium possess these enzymes [43,44,45]. The energy-consuming futile cycles composed of reactions catalyzed by these two enzymes and by corresponding partners, such as 6-phosphofructokinase or pyruvate kinase reactions [46,47,48,49], can affect the central metabolism of the cells [50]. Two alternative pathways for acetate synthesis were considered: first, acetate synthesis from acetyl-CoA via reversible reactions of the phosphate acetyltransferase and acetate kinase and, second, acetate synthesis from pyruvate via an irreversible pyruvate oxidase reaction. Both pathways are known to be active in *E*. *coli* [51,52,53]. Accounting for both pathways for acetate synthesis may affect the accuracy of the pyruvate dehydrogenase (PDH) flux estimation. Thus, the PDH flux was characterized by an interval in which the lower boundary was limited by the acetyl-CoA requirement for biomass synthesis and the upper boundary was determined under the assumption that all secreted acetate is synthesized from acetyl-CoA.

To account for CO_2_-associated carbon transfer, reactions accompanied by CO_2_ production or consumption were expressed in an explicit manner including an anabolic reaction and a reaction of CO_2_ exchange with an environment modeled as specified in [54].

Two known pathways for the glycine synthesis in *E*. *coli*, from serine and threonine [55], were included into the model. According to the previously performed analysis of cells grown aerobically on ^13^C-labeled glucose, the glycine cleavage is irreversible [56].

The reversible reactions were modeled as described in [57], that is, as forward (F) and reverse (R) fluxes, the difference between which gives a value of net flux through the reversible reaction.

The amino acid biosynthesis reactions, data on the mass isotopomer distribution (MID) of which were used for flux calculation, were explicitly expressed. To account for carbon transfer associated with biomass synthesis, reactions of nucleotides biosynthesis were explicitly expressed as well. One example is carbon transfer from the aspartate pool to the fumarate pool when aspartate is used as a donor of the amino group. Metabolites drained for biomass synthesis were accounted for by a single biomass equation, as described in Section 2.9.4.

Atom transition schemes were extracted from the literature [58]. The measured external carbon fluxes (effluxes) were biomass synthesis, efflux of secreted acetate, and the glucose uptake rate.

#### 2.9.4. Requirements for Biomass Synthesis

As ^13^C-MFA is based on both carbon and isotopomer balancing, accounting for precursors draining from the central metabolism for biomass synthesis is crucial for carbon flux estimation [59,60,61]. To account for the consumption of precursors and energy for biomass synthesis, a biomass equation is usually formulated based on the macromolecular composition of the cellular biomass and the need for precursors to synthesize the building blocks of each macromolecule [62]. The flux to biomass synthesis is equal to a specific growth rate (1/h).

DNA, RNA, protein (Prot), peptidoglycan (PGL), phospholipids (PLP), lipopolysaccharides (LPS), and glycogen (GL) constitute 96% of the dry weight of *E*. *coli* cells [63]. As protein and RNA are the most abundant and variable biomass components [59], they were measured for each strain, as described in Section 2.9.5. The contents of the other biomass components were taken from the literature [63], as described below. The requirements for DNA, RNA, and protein synthesis were expressed as draining of the corresponding building blocks—dNTP, NTP, and amino acids. The DNA, RNA, and protein compositions were taken from the literature [63]. The precursor requirements for the synthesis of other cellular components were recovered from biosynthetic pathways described in the EcoCyc database (www.ecocyc.org, accessed on 31 October 2019).

A stoichiometric coefficient in the biomass equation denotes an amount of a constituent drained for the synthesis of 1 g of biomass (mmol/g_DW_) [64,65]. The dry weight (DW) of the strains was determined by multiplying the measured OD_595_ value by the correlation coefficient between OD_595_ and DW of the cells (mg of the cells in 1 mL of a cell culture at OD_595_ of 1). To determine this coefficient, cells from 3–5 mL of exponentially growing culture were placed onto a pre-weighed 0.45 μm filter (MF-Millipore™ Membrane Filter, Merck,), followed by drying at 60 °C until reaching a constant weight.

DNA is a slightly variable component [59]. Therefore, its content in dry weight for all of the tested strains was assumed to be the same, and it was set to 3.1% (g_DNA_/g_DW_) [63]. The contents of the other biomass components were approximated using the values known from the literature [63] to account for 100% of dry weight (Equation (2)) and through their mass ratios from the literature (Equation (3)):PLP + LPS + PGL + GL = DW − (Prot + RNA + DNA)(2)
and
PLP/LPS/PGL/GL = 9.1:3.4:2.5:2.5(3)

The biomass compositions of the *E*. *coli* strains that were used are shown in Appendix A.

#### 2.9.5. Determination of Protein and RNA Content

Biomass samples for the estimation of the protein and RNA content were collected by centrifugation, washed with 0.9% NaCl, and stored at −70 °C before use.

The protein content in the biomass was estimated using a Bio-Rad DC Protein Assay kit (Bio-Rad, Hercules, CA, USA) according to the manufacturer’s instructions.

To separate two nucleic acids, RNA and DNA, the Schmidt−Tannhauser method was used [66]. At first, biomass samples were incubated with ice-cold 0.25 N HClO_4_ (1 mL) at 0 °C for 30 min to remove the low-weight cytoplasmic metabolites. During incubation, the test tubes containing the samples were gently mixed 2–3 times. Then, the biomass was collected at 8000× *g* for 3 min and incubated in 0.5 mL of 1 N KOH at 37 °C for 1 h, to hydrolyse the RNA to the monomers. The test tubes were mixed 2–3 times using a vortex. Then, the hydrolysed RNA was separated from the protein and DNA by adding ice-cold 3N HClO_4_ (0.5 mL) followed by centrifugation at 13,000× *g* for 5 min at 0 °C. The UV spectra of the obtained supernatants were recorded using a NanoDrop 2000 spectrophotometer (Thermo Scientific, Waltham, MA, USA). The RNA concentration (g/L) in the solution was calculated using the following formula [67,68]:(4)$$C=\left(OD_{270}-OD_{290}\right)\times \frac{10.3}{0.19}$$
where 0.19 is value of the (*OD*_270_−*OD*_290_) difference, which corresponds to nucleic acid hydrolysate with a nucleic acid phosphate concentration of 1 mg/L, and 10.3 is the average coefficient to transfer the phosphate amount to the ribonucleotides amount.

#### 2.9.6. GC-MS Analysis

Amino acids extracted from protein, ribose from RNA, and glucose from the glycogen of the biomass were utilized as sources of data regarding mass-isotopomer abundance.

Before the analysis, the protein, RNA, and glycogen of the biomass were hydrolyzed under acidic conditions, down to corresponding monomers. The derived amino acids, ribose, and glucose were modified by derivatization prior to separation by gas chromatography. The acid hydrolysis of the cell protein was carried out in sealed test tubes (Vacuum Hydrolysis test tube, Thermo Scientific, USA) according to the method found in the literature [69]. Released proteinogenic amino acids were then silylated by N-*tert*-butyldimethylsilyl-N-methyltrifluoroacetamide (TBDMSTFA) with 1% tertbutyldimethylchlorosilane (TBDMST) (Merck, Germany) treatment according to a previously developed method [69], followed by GC-MS analysis.

Ribose and glucose aldononitrile propionate derivatives were prepared according to the previously established method [70].

Gas chromatography−mass spectrometry (GC−MS) analysis was performed using an Agilent 7890 B gas chromatograph equipped with a DB-5MS capillary column (30 m, 0.25 mm i.d., and 0.25 μm-phase thickness) connected to an Agilent 5977A mass spectrometer operating under electron ionization (EI) at 70 eV. The GC−MS analysis of the amino acid derivatives was based on the method found in the literature [69]. First, 1 µL of the sample was injected at a 1:10 split ratio. Amino acid derivatives were detected in scan mode from 140 to 550 m/z.

The GC−MS analysis of sugar derivatives was based on the method from the literature [70]. Here, 1 µL of the sample was injected at 1:2 to 1:10 split ratios. The m/z 173 and 284 fragments of the ribose derivative, and the m/z 173 and 370 fragments of the glucose derivative were measured in single ion monitoring (SIM) mode.

The chromatograms were analyzed using the MassHunter Workstation Software. Calculation of the mass isotopomer distributions (MIDs) of the amino acid, ribose, and glucose fragments was carried out based on the peak areas of the molecular ion fragments [71].

The MIDs of two fragments of derivatized ribose (Rib173 and Rib284) and two fragments of derivatized glucose (Glc173 and Glc370), in addition to the MIDs of 31 fragments of amino acid TBDMS-derivatives (Ala232, Ala260, Val260, Val288, Leu274, Ile200, Ile274, Phe302, Phe308, Phe336, Met218, Met292, Met320, Gly218, Gly246, Ser288, Ser390, Thr376, Thr404, Tyr302, Tyr438, Tyr466, Tyr508, Asp302, Asp376, Asp390, Asp418, Glu330, Glu404, and Glu432), were utilized as the experimental data for the metabolic flux calculation.

#### 2.9.7. Flux Calculation and Statistics Analysis

Flux calculation and statistics analysis were carried out using a modified version of the previously developed OpenFLUX2 software [72], adapted for the high-throughput flux estimation of middle-sized metabolic models in both single- and parallel-labeling experiment settings. The main change in the OpenFLUX2 software was the development and integration of a special optimization engine, which allowed for the accelerated calculation of the elementary-metabolite-unit (EMU)-based metabolic model. The core of the optimization engine includes graph-theoretic algorithms, which perform modifications on the structure of the EMU-based model. The following procedures should be highlighted: EMU model graph decomposition on the basis of strongly connected components and the detection of isomorphic EMU subgraphs. Furthermore, the optimization engine provides technical improvements to the EMU model calculation, such as optimization of the EMU equation calculations by using sparse matrix representations and other memory allocation techniques. The intracellular fluxes were calculated by solving the weighted regression problem, and involve the minimization of the variance-weighted sum of squared residuals (SSRs) between experimentally measured data (GC−MS-detected mass-isotopomer distributions and extracellular fluxes) and model-generated data. Random repeats were used to account for the non-convex nature of the EMU model and the consequent non-linearity of the regression problem. First, 300 independent points were sampled from the feasible free flux space. Flux parameters were then estimated for every sampled point and, finally, the flux parameters with minimal SSRs that passed the χ^2^-test were selected as the optimal solution for the given metabolic model.

The uncertainty of the estimated flux parameters was analyzed by the parametric bootstrap confidence intervals (PB-*CI*s) [73], which were calculated using the percentile method [74,75]. This method was previously implemented in OpenFLUX2 and is known as the “discarding” method of the Monte Carlo simulations [72]. The settings of the confidence intervals estimation procedure were adjusted so that all confidence−interval borders for the net fluxes of interest reached convergence.

## 3. Results

### 3.1. Heterologous Expression of H^+^-PPase^Rru^ in E. coli with Replacement of the Native S-PPase

The codon-harmonized gene *hppa*^Rru^ encoding H^+^-PPase^Rru^ from *R. rubrum ATCC 11,170* was chemically synthesized and cloned into the integrative vector pAH162-Tc^R^-2Ter [31]. The resulting recombinant plasmid pAH162-Tc^R^-2Ter-*hppa*^Rru^ carrying the promoter-less *hppa*^Rru^ gene was integrated into the artificial *IS*5.8::ϕ80-*attB* site of the *E*. *coli* chromosome. The expression of the gene *hppa*^Rru^ was then activated in the *E*. *coli* chromosome by insertion of the “strong” constitutive λP_L_ promoter upstream the gene via λRed-mediated recombination (for details, see Material and Methods).

To assess the biological functionality of individual heterologous *R. rubrum* H^+^-PPase^Rru^ as the sole inorganic pyrophosphatase in *E. coli* cells, we attempted to inactivate the native, cytoplasmic pyrophosphatase PPase. The corresponding gene *ppa* is essential and cannot be inactivated in *E. coli* of the wild-type under physiological conditions [27]. Moreover, this chromosomal deletion could not be obtained in the present study on the basis of *E. coli* strain with the integrated promoter-less variant of the *hppa*^Rru^ gene as well (data not shown). However, the presence of the λP_L_ promoter upstream the harmonized H^+^-PPase^Rru^ gene allowed us to delete the *E. coli ppa* gene and thereby compensate for the loss of native S-PPase activity in the recombinant strain. The obtained *ppa*-deficient recombinant clones possessed a slow-growing phenotype on the LB medium compared with the wild-type.

Emergence of *ppa* gene deletion due to λRed-driven integration of an artificial linear DNA fragment marked by the excisable Cm^R^-gene in the recombinant *E. coli* chromosome was initially unambiguously confirmed by PCR (see, Appendix D). Moreover, the specially provided Southern blot hybridization [28] of the biotin-labeled probes containing an encoding part of the *ppa* gene verified that this fragment had no homologous loci anywhere in the obtained recombinant *E. coli* genome (Figure 2). For the Southern blot analysis, *E. coli* chromosomal DNA was independently digested with the two restriction enzymes, *EcoR*I and *Hind*III, and hybridized with a *ppa*-carrying DNA fragment. The autoradiogram from the Southern blot hybridization of DNA from the selected isolates showed the presence of hybridization of two different chromosome restriction probes with the specific DNA fragment of *ppa* gene in case of the wild-type strain (lanes 1 and 3, Figure 2), whereas no hybridization with a specific DNA fragment of *ppa* was observed in the case of chromosome restriction probes of the strain harboring replacement of native S-PPase with H^+^-PPase^Rru^ one (lanes 2 and 4, Figure 2). Thus, the possibility of replacing the native essential S-PPase in *E. coli* cells with the membrane H^+^-PPase^Rru^ was demonstrated for the first time.

### 3.2. Analysis of the Growth of Strains Containing H^+^-PPase^Rru^ and the Effects of Increasing hppa^Rru^ Gene Copy Number

Analysis of the H^+^-PPase^Rru^-containing strain growth confirmed the in vivo biological functionality of H^+^-PPase^Rru^ as a sole inorganic pyrophosphatase in *E. coli* cells. Despite *ppa* gene deletion reducing the growth rate of the strain by almost 45% on M9 minimal medium when the replacing native S-PPase with H^+^-PPase^Rru^ (Figure 3), H^+^-PPase^Rru^ was nevertheless still able to compensate for the absence of native S-PPase. Moreover, in this case, PP_i_ cleavage should be accompanied by proton translocation across the membrane, in contrast with S-PPase. The decreased growth rate of the strain with the replacement of *E. coli* PPase with the membrane H^+^-PPase^Rru^ can be explained by data on the kinetic properties of PPases from the literature. In terms of catalysis, M-PPases are the slowest (k_cat_~10 s^−1^) of the PPases, whereas the turnover of family I PPases, of which the PPase from *E. coli* belongs, is one order of magnitude higher (k_cat_~200 s^−1^) [76].

In the attempt to compensate for the low catalytic rate of M-PPases, we constructed the strains carrying two and three copies of the *hppa*^Rru^ gene under ∆*ppa* genetic background. Each of the P_L_-*hppa*^Rru^ cassette integrated into alternative loci of the *E. coli* chromosome restored the growth of the *ppa*-deficient mutant to almost the same level as the original copy at the *IS*5.8 locus (Appendix A). However, additional copies did not increase cell growth and retained it at the lower level in comparison with the wild-type *E. coli* strain (Figure 4). Various reasons for this can be assumed: disbalance between the high-efficient transcription of the gene from the strong promoter and non-efficient translation/secretion of its membrane-bound protein product, locus-dependent negative effect of the insertion(s) itself, or the low capacity of M-PPase to generate proton motive force under certain experimental conditions. The involvement of all of these possibilities requires further thorough investigation for elucidation. It has to be noted, as well, that while the presence of one copy of the integrated *hppa*^Rru^ gene in the chromosome of wild-type cells carrying the native *ppa*-gene practically did not change the growth rate of bacteria (Figure 3), increasing the number of *hppa*^Rru^ gene copies up to two and three led to some decrease, by 10–20%, in the growth rate of the resulted strains (data not shown).

### 3.3. Analysis of PPase Activity of E. coli Strains Containing S- and M-PPases

To support the idea about the possibility of replacing the soluble inorganic PPase with the membrane-bound PPase and to study the functioning of H^+^-PPase^Rru^ as a sole inorganic PPase in *E. coli* cells, known methods for PPase activity measurement were adapted for analyzing the PPase activity of both types of this enzyme. PPase activity in the soluble fraction of the strains possessing S- and/or M-PPases was measured in accordance with the previously described method [40]. As expected, the measurement of the PPase activity in the strain with the expression of heterologous H^+^-PPase^Rru^ and deletion of the native *ppa* gene encoding S-PPase showed a decrease in PPase activity in the soluble fraction to undetectable levels (Table 3).

For the detection of the PPase activity in the membrane fraction, inverted membrane vesicles (IMVs) were isolated [77]. The reaction was initiated by adding 0.1 mM PP_i_ and 2 mM Mg^2+^, followed by a fluorometric assay, which is quite sensitive and capable of detecting a low P_i_-releasing enzymatic activity. In this experiment, the undesirable activity of soluble pyrophosphatase that could be present in the preparations of IMVs was suppressed by adding 0.5 mM NaF to the reaction mixture. The activity of soluble PPase is blocked by this inhibitor, whereas H^+^-PPase^Rru^ is fluoride-resistant [77].

The analysis of the time course of IMV-associated phosphate production (Figure 5) showed that the membrane pyrophosphatase activity in the IMVs of the strain where native S-PPase was replaced with H^+^-PPase^Rru^ was about 10-fold higher compared with the IMV-associated activity in the wild-type strain (Figure 5). Thus, the obtained result supports the assumption that H^+^-PPase^Rru^ is capable of acting as a sole PP_i_-hydrolyzing enzyme in *E. coli*. Interestingly, an unexplainably low rate of pyrophosphatase activity was detected in IMVs of the strain containing both types PPases, soluble and membrane-bound. At present, the reason underlying this effect is not clear and requires further elucidation. For the wild-type strain MG1655, a small but detectable PPase activity was observed. This indicates that the possibility of contamination of IMV preparations with S-PPase cannot be completely excluded in these experiments. Similar contamination was observed in the work of Belogurov et al., where the heterologous expression of fully functional H^+^-PPase^Rru^ in *E. coli* cells was reported for the first time [10].

### 3.4. Carbon Flux Distribution in E. coli MG1655, MG1655 IS5.8::P_L_-hppa^Rru^ and MG1655 IS5.8::P_L_-hppa^Rru^ ∆ppa::cat Strains

Taking into account the different properties and physiological roles of soluble and membrane PPases, in particular, the ability of the latter to generate a proton electrochemical gradient, estimating the effect of its expression on the fluxome of the wild-type *E. coli* strain was of interest. To this end, the ^13^C-MFA of strains MG1655, MG1655 *IS*5.8::P_L_-*hppa*^Rru^, and MG1655 *IS*5.8::P_L_-*hppa*^Rru^ ∆*ppa*::*cat* was performed. In the first step, the strains were cultivated (see Section 2.9.1) on naturally labeled glucose to confirm steady-state growth (Appendix A) and to determine the growth parameters (Appendix A). The strains MG1655 and MG1655 *IS*5.8::P_L_-*hppa*^Rru^ have the same growth parameters. However, the strain MG1655 *IS*5.8::P_L_-*hppa*^Rru^ ∆*ppa*::*cat* in addition to slower growth, mentioned above, possesses a lower specific glucose consumption rate and is characterized by a statistically significant decrease in the biomass yield on glucose, as well as an increase in acetate yield on glucose (see Appendix A). This could indicate the re-distribution of internal carbon fluxes in this strain. At the end of cultivation, the samples were taken to determine the protein and RNA content (see Section 2.9.5) and the correlation coefficient between OD_595_ and DW (K_DW/OD_, see Section 2.9.4). The coefficient K_DW/OD_ for the strain MG1655 *IS*5.8::P_L_-*hppa*^Rru^ ∆*ppa*::*cat* was almost 1.3 times higher than that of strains MG1655 and MG1655 *IS*5.8::P_L_-*hppa*^Rru^. A rather complicated relationship between cell size, volume, dry weight and optical density was mentioned in the literature [78,79]. Thus, the observed difference in K_DW/OD_ values could be explained by the difference in cell form/size of the strains. Indeed, microscopic analysis revealed that cells of the MG1655 *IS*5.8::P_L_-*hppa*^Rru^ ∆*ppa*::*cat* strain have a more elongated form. Although a more detailed analysis of cells parameters could be interesting in itself, but it was beyond the scope of this publication. Then, a carbon-labeling experiment was carried out using 100% [1,2-^13^C]glucose as a labeled substrate, which was recognized as one of the most effective labeled substrates for overview of *E. coli* metabolism when the single labeling experiment is carried out [80]. Finally, the mass isotopomer distribution of proteinogenic AAs, ribose from RNA, and glucose from glycogen were measured by GC−MS (see Section 2.9.6). Optimal flux parameters and their confidence intervals were estimated as described in Section 2.9.7.

Carbon flux distribution in the upper part of the central carbon metabolism of *E*. *coli* MG1655 strain (Figure 6) correlates well with the previously published data [81]. The differences observed in the lower part of the metabolic map (Figure 6) may be caused by more sensitivity of the corresponding fluxes to the cultivation conditions (for example, aeration).

The carbon flux distribution of the strain MG1655 *IS*5.8::P_L_*-hppa*^Rru^, carrying both native S-PPase and heterologous H^+^-PPase^Rru^, is similar to that of the wild-type strain MG1655 (Figure 7). This coincides with the low PPase activity measured in the membrane fraction of this strain. Probably, the H^+^-PPase^Rru^ characterized by a slower turnover cannot compete with native S-PPase for PP_i_.

The *E*. *coli* strain where native S-PPase was replaced with membrane M-PPase is characterized by a statistically significant 36% decrease in the portion of carbon utilized via TCA cycle compared with the wild-type *E*. *coli* strain MG1655 (for comparison, see Figure 6 and Figure 8). One can suppose that this may be a consequence of the decrease in precursors’ demand for biomass synthesis, as the strain MG1655 *IS*5.8::P_L_-*hppa*^Rru^ ∆*ppa*::*cat* possesses a reduced biomass yield. However, the biomass yield is reduced by only 23%. The main source of energy in aerobically grown cells is the TCA cycle [83]. One molecule of ATP and two molecules of NADH, which can then be oxidized in the electron transport chain to provide ATP, are generated in the cycle. Additionally, the succinate dehydrogenase reaction directly donates electrons to the electron transport chain. Hence, the significant (about 36%) decrease in the TCA cycle carbon flux in *E*. *coli* MG1655 with the replacement of native S-PPase with the membrane M-PPase unambiguously confirms the considerable increase in the production of energy—for example, as ATP—by an alternative route in this strain. The carbon that is not used in energy generation is redirected to acetate synthesis. At the same time, a slight decrease in oxidative pentosophosphate (PP) pathway flux was detected in the MG1655 *IS*5.8::P_L_-*hppa*^Rru^ ∆*ppa::*Cm^R^ strain compared with the MG1655 strain. Together with the decrease in TCA flux, this should lead to an increase in a portion of NADPH generated by H^+^-dependent PntAB transhydrogenase to supply enough NADPH for biomass biosynthesis [84]. Although NADPH balancing revealed an increase in the amount of NADPH produced by PntAB transhydrogenase, it was not statistically significant (data not shown).

^13^C-MFA is based on a data set, including both MIDs and effluxes values. The letter accounts, also, for drain of metabolites from central metabolism for biomass synthesis, which is expressed in the metabolic model as a single biomass equation (see Section 2.9.4). Taking into account that the coefficient K_DW/OD_ influences stoichiometric coefficients of this equation and that value of this coefficient for strain MG1655 *IS*5.8::P_L_-*hppa*^Rru^ ∆*ppa::*Cm^R^ differs from that of the other analyzed strain (which have almost similar values of the coefficient K_DW/OD_), we repeated flux calculation for strain MG1655 *IS*5.8::P_L_-*hppa*^Rru^ ∆*ppa::*Cm^R^ assuming a K_DW/OD_ value similar to that of the MG1655 strain, i.e., K_DW/OD_ = 0.51. Changing the K_DW/OD_ value indeed affected the biomass composition (see Appendix A) and, as a consequence, the stoichiometric coefficients of the biomass equation. However, the observed flux re-distribution effect, in particular a decrease in TCA fluxes, in strain MG1655 *IS*5.8::P_L_-*hppa*^Rru^ ∆*ppa::*Cm^R^ compared to strains MG1655 and MG1655 *IS*5.8::P_L_*-hppa*^Rru^, was not affected.

The flux re-distribution, observed by ^13^C-MFA, confirmed that the cells where native S-PPase was replaced with H^+^-PPase^Rru^ had a different metabolic state, which served as a basis for testing the use of H^+^-PPase^Rru^ for increasing the production of substances whose synthesis require high levels of energy.

## 4. Discussion

In the present study, *E. coli* cells possessing only the soluble cytoplasmic form of PPase were engineered for the heterologous expression of membrane-bound H^+^-PPase from *R. rubrum.* For this purpose, the gene coding for H^+^-PPase^Rru^ was chemically synthesized with codons optimized for the expression in *E. coli* cells*;* codons were harmonized [34], i.e., altered to maintain the frequency of their usage in *E. coli* similar to that in *R. rubrum*.

Here, to prove in vivo biological functionality of H^+^-PPase^Rru^ as a sole inorganic pyrophosphatase in *E. coli* cells, replacement of the native, normally essential [27], soluble cytoplasmic *E. coli* PPase with the membrane-bound H^+^-PPase^Rru^ was demonstrated for the first time. The ability of H^+^-PPase^Rru^ to act as the sole PP_i_-hydrolyzing enzyme in *E. coli* cells was confirmed genetically and biochemically. Various expression cassettes harboring the *hppa*^Rru^ gene under the control of various promoters, λP_L_, P_tac_, and P*_tac_* derived with a lower affinity to RNA polymerase were tested, and only the “strongest” promoter (λP_L_) allowed for cell growth of the *ppa*-deficient strain (data not shown). Notably, even in this case, the specific growth rate was 45% lower for these cells compared with the wild-type *E. coli* strain carrying the native PPase. At the same time, in that case, the expression of H^+^-PPase^Rru^ itself had no negative effect on growth.

As shown earlier, heterologous H^+^-PPases from *Arabidopsis thaliana* and *Chloroflexus aurantiacus* can complement in vivo a low level of one of the soluble PPases (*IPP1*), which is essential for the yeast *Saccharomyces cerevisiae* [85]. However, complementation of the absence of bacterial soluble cytoplasmic PPase has not been previously demonstrated, and the obtained *ppa*-deficient strains heterologously expressed H^+^-PPase^Rru^ can be a useful tool to study further the structure–function relationships in this class of proton pumps.

Nyre’n and Strid [86] hypothesized that the main function of the H^+^-PPase of *R. rubrum* is to maintain the proton motive force in light-grown cells under conditions of low energy. Later, García-Contreras et al. presented experimental evidence, indicating that H^+^-PPase of *R. rubrum* provides this photosynthetic bacterium with an alternative energy source that is important for growth in low-energy states [87].

To support the hypothesis of the bioenergetic advantages of membrane-bound H^+^-PPase^Rru^ expression in *E. coli* cells, ^13^C-MFA was applied. ^13^C-MFA confirmed that replacing native PPase with H^+^-PPase^Rru^ leads to carbon flux re-distribution. The main metabolic changes induced by the replacement of native *E. coli* PPase with membrane-bound H^+^-PPase^Rru^ include decreased carbon fluxes in the tricarboxylic acid cycle (TCA) and pentose phosphate pathway (PPP) and increased acetate synthesis. The magnitude of these changes are 1.5 times higher than the magnitude of changes in the biomass yield, which can potentially lead to flux re-distribution, too, as less drain of metabolites from the central metabolism for biomass biosynthesis is required. In addition, the observed flux re-distribution is in good agreement with the assumption that the alternative energy pathway provided by H^+^-PPase^Rru^ results in less carbon outflow for energy generation by traditional *E. coli* pathways such as TCA cycle. Despite significant progress in elucidating the unique properties of M-PPases in recent years, further study of these enzymes seems important for evaluating their biotechnological potential and expanding their scope of application.

## 5. Conclusions

In this study, we investigated the possibility of the H^+^-translocating membrane-bound pyrophosphatase (H^+^-PPase) from *Rhodospirillum rubrum* to fuel *Escherichia coli* cells via an alternative pathway for energy generation.

The obtained data allow for hypothesizing novel applications of the expression of proton-pumping membrane-bound PPases for the microbial production of a number of natural and artificial compounds whose biosynthesis is dependent on ATP consumption, for instance, proteins, antibiotics, nucleosides, and amino acids and their derivatives.

## 6. Patents

Malykh E., Stoynova N., Golubeva L., Mashko S., Kovaleva E., Shupletsov M., Baboshin M., Krylov A., Zakataeva N., Andrianova E., Kharchenko M., Voroshilova E. Method for producing target substance by bacterial fermentation. Patent application WO2020071538A1.

## Figures and Tables

**Figure 1 microorganisms-11-00294-f001:**
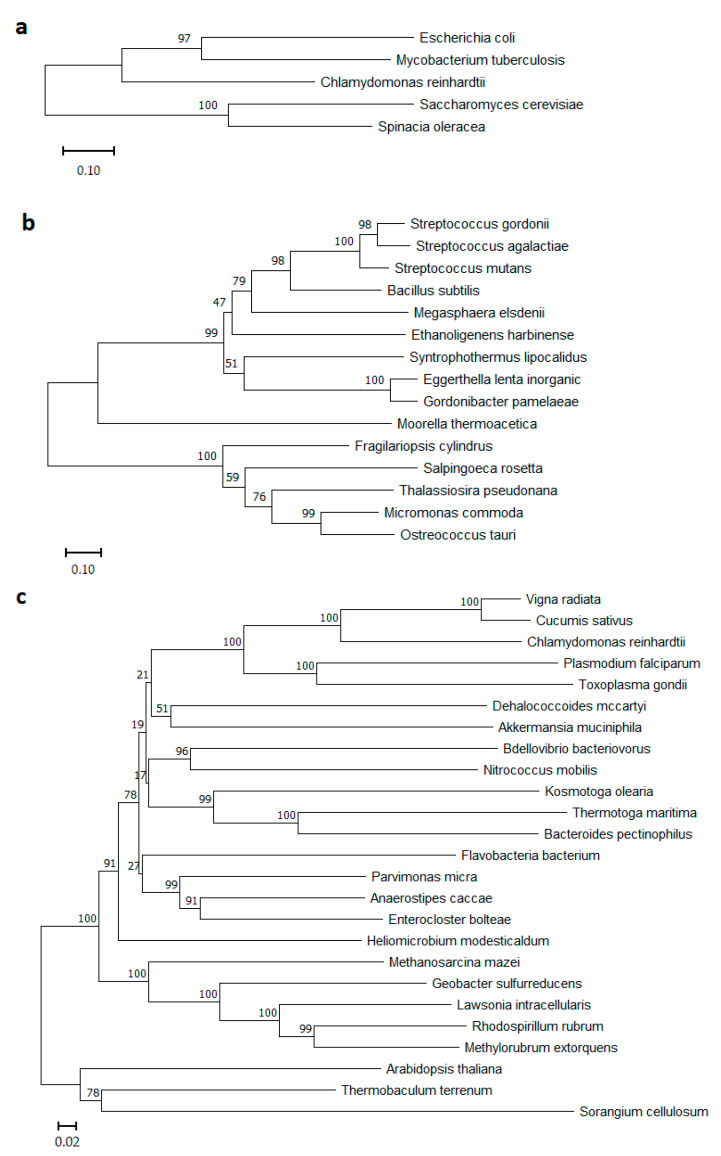
Phylogenetic trees of (**a**) soluble PPases Family I, (**b**) soluble PPases Family II, and (**c**) membrane-bound PPases protein sequences. Phylograms were generated using NJ analysis (MEGA software, https://www.megasoftware.net/, accessed on 23 December 2022), and multiple sequence alignment was performed by Muscle [3]. Bootstrap values of 1000 replicates are shown in percentages at the internodes. Scale bars 0.1 for (**a**,**b**), and 0.02 for (**c**) substitutions per residue are indicated.

**Figure 2 microorganisms-11-00294-f002:**
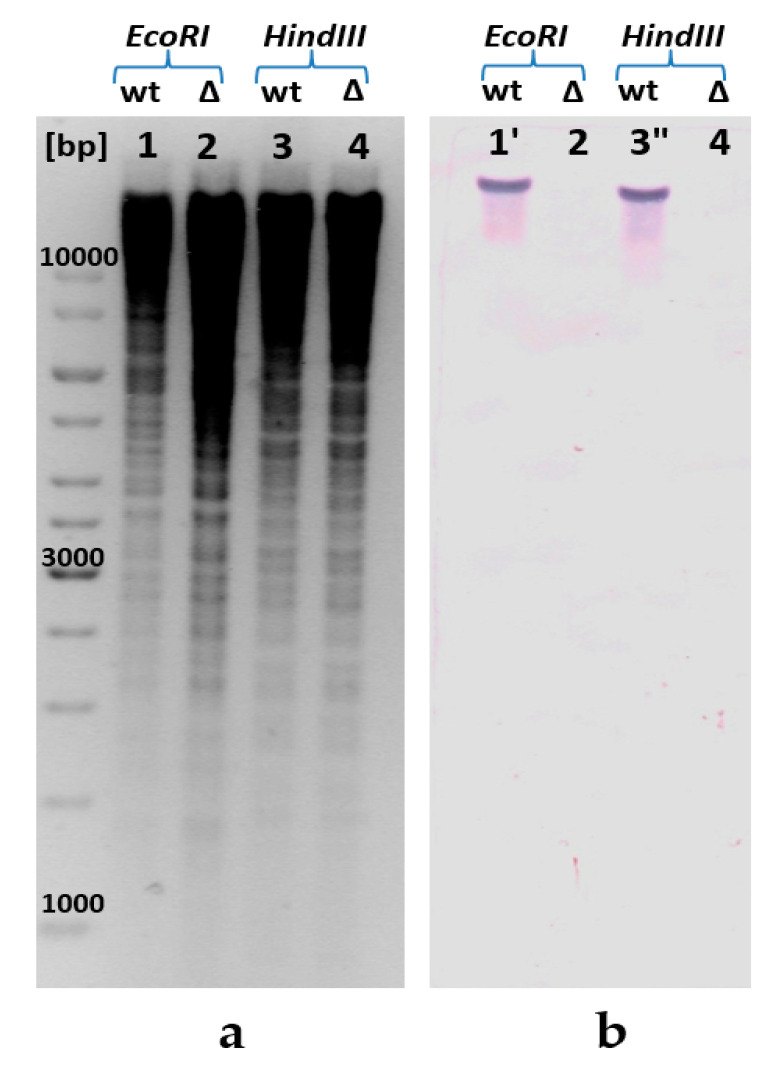
Confirmation of the presence of *ppa* deletion in the *E. coli* chromosome by Southern blot analysis: (**a**) for the Southern blot analysis, genomic DNA was digested with *EcoR*I (lanes 1, 2) and *Hind*III (lanes 3, 4), and (**b**) hybridized with a *ppa*-carrying DNA fragment amplified by PCR. The calculated size of the *ppa*-carrying *EcoR*I-fragment is 18,447 bp (1′); calculated size of *ppa*-carrying *Hind*III fragment is 16,618 bp (3′′). [bp], DNA ladder; wt, wild-type MG1655; ∆—MG1655 *IS*5.8-P_L_-*hppa*^Rru^ ∆*ppa*::*cat*.

**Figure 3 microorganisms-11-00294-f003:**
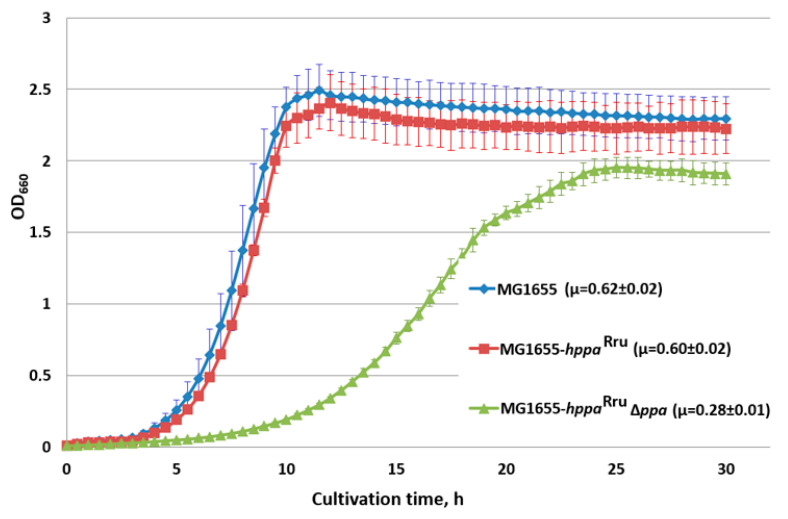
Growth of strains with the *hppa*^Rru^ gene in the presence and absence of *E. coli ppa*. Cells were grown with aeration in L-tubes with M9 minimal medium supplemented with 0.3% of glucose at 37 °C in an Advantec photorecorder. MG1655, MG1655 wild-type; MG1655-*hppa*^Rru^, MG1655 *IS*5.8::P_L_-*hppa^Rru^*; MG1655- *hppa*^Rru^ ∆*ppa*, MG1655 *IS*5.8::P_L_-*hppa*^Rru^ ∆*ppa*. Average data are shown; bars refer to standard deviations from three independent experiments.

**Figure 4 microorganisms-11-00294-f004:**
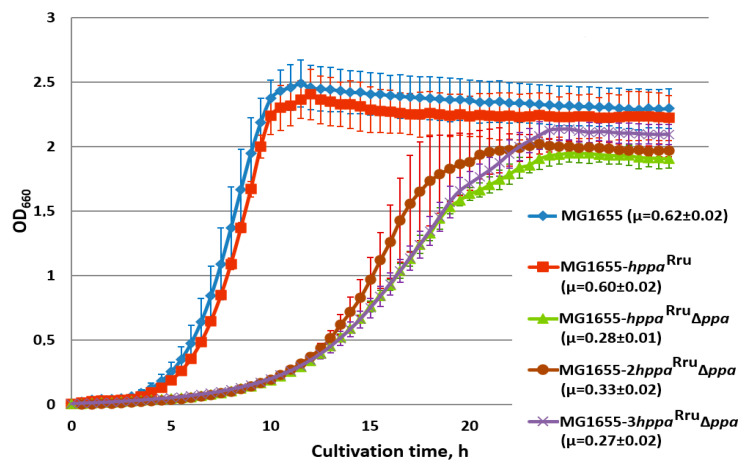
Growth of strains, harboring 1, 2, or 3 copies of the *hppa*^Rru^ gene in the presence and absence of *E. coli ppa*. Cells were grown with aeration in L-tubes with M9 minimal medium supplemented with 0.3% of glucose at 37 °C in an Advantec photorecorder. MG1655, MG1655 wild-type; MG1655-*hppa*^Rru^, MG1655 *IS*5.8::P_L_-*hppa*^Rru^; MG1655-*hppa*^Rru^ ∆*ppa*, MG1655 *IS*5.8::P_L_-*hppa*^Rru^ ∆*ppa*; MG1655-2*hppa*^Rru^ ∆*ppa*, MG1655 *IS*5.8::P_L_-*hppa*^Rru^ *adrA*::P_L_-*hppa*^Rru^ ∆*ppa*; MG1655-3*hppa* ∆*ppa*, MG1655 *IS*5.8::P_L_-*hppa*^Rru^ *adrA*::P_L_-*hppa*^Rru^ *∆adhE*::P_L_-*hppa*^Rru^ ∆*ppa*. Average data are shown; bars refer to standard deviations from three independent experiments.

**Figure 5 microorganisms-11-00294-f005:**
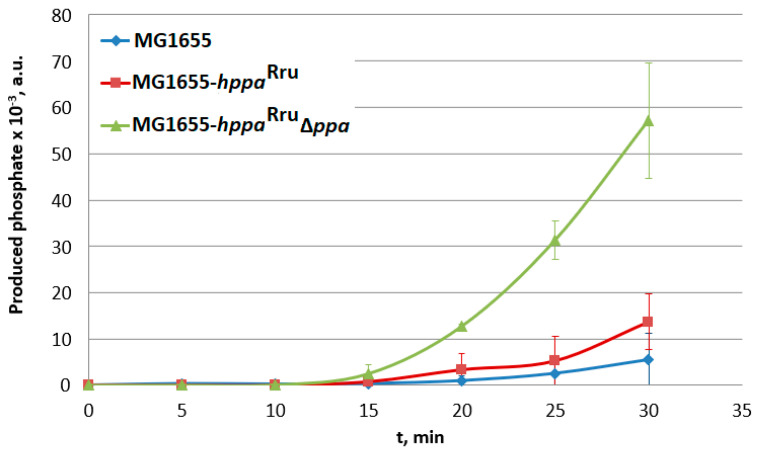
Analysis of PPase activity in inverted membrane vesicles (IMVs) of the strains containing cytoplasmic and/or membrane PPases. MG1655-*hppa*^Rru^, MG1655 *IS*5.8::P_L_-*hppa*^Rru^; MG1655-*hppa*^Rru^ ∆*ppa*, and MG1655 *IS*5.8::P_L_-*hppa*^Rru^ ∆*ppa*::*cat.* Average data are shown; bars refer to standard deviations from three independent measurements; all experiments were performed with one IMV preparation.

**Figure 6 microorganisms-11-00294-f006:**
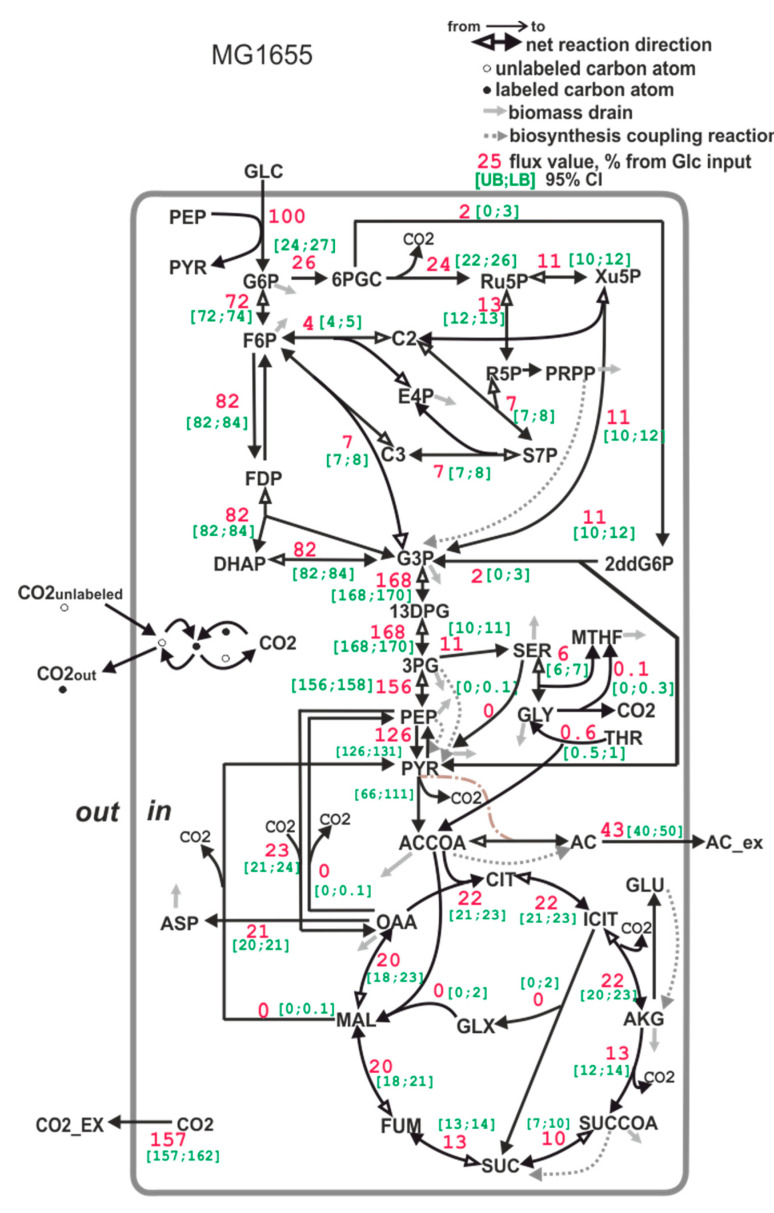
The carbon flux distribution in the *E*. *coli* MG1655 strain. Flux through the PEP- > PYR reaction is a sum of the PTS-dependent glucose transport reaction and pyruvate kinase reaction. Flux through the pyruvate dehydrogenase reaction is expressed as a range (see Section 2.9.3.) Accounting for serine degradation to pyruvate (activated, for example, in an *E. coli pfkA*-deficient mutant, see [82]) leads to a broad confidence interval of lumped fluxes of phosphoglycerate mutase and enolase, and of pyruvate kinase flux. All of the fluxes are normalized to a specific glucose uptake rate (mmol/gDW*hour), which is set to 100%. The [UB;LB] 95% CI in the legend is the [Upper bound (UB); Lower bound (LB)] of the 95% confidence interval, calculated as described in Section 2.9.7.

**Figure 7 microorganisms-11-00294-f007:**
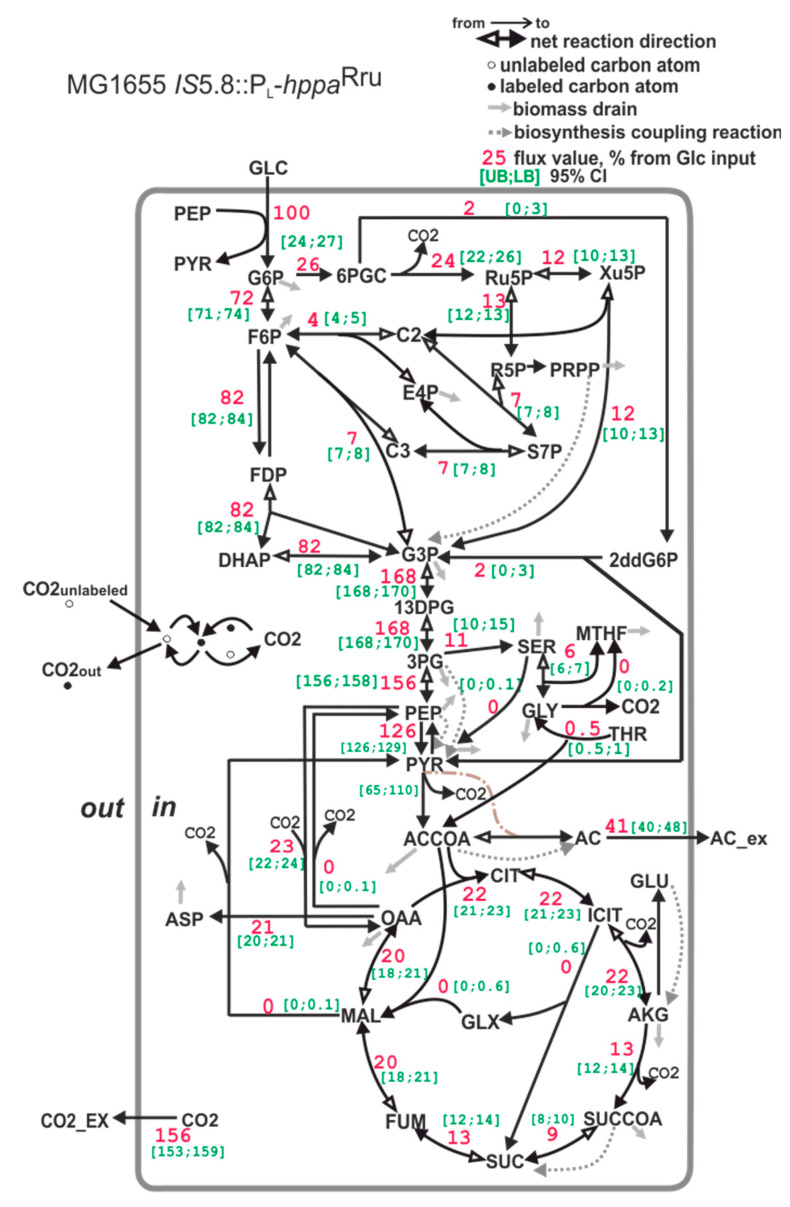
The carbon flux distribution in the *E*. *coli* MG1655 *IS*5.8::P_L_-*hppa*^Rru^ strain. For details, see the caption of Figure 6.

**Figure 8 microorganisms-11-00294-f008:**
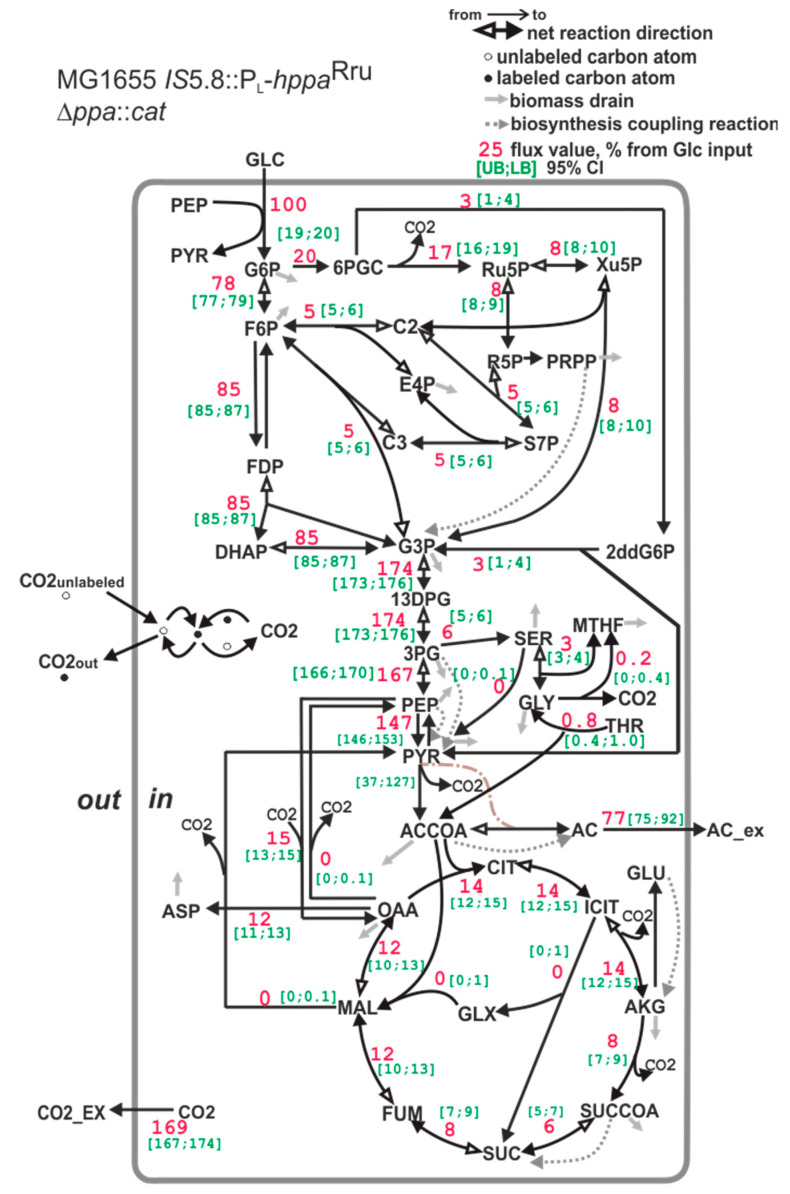
The carbon flux distribution in the *E*. *coli* MG1655 *IS*5.8::P_L_-*hppa*^Rru^ ∆*ppa::*Cm^R^ strain. For details, see the caption of Figure 6.

**Table 1 microorganisms-11-00294-t001:** Bacterial strains and plasmids used in this study.

Strain or Plasmid	Description	Reference or Source
MG1655	*Escherichia coli* K12 wild-type	VKPM ^a^ B6195
BL21 (DE3)	*E. coli* B F^−^ *ompT gal dcm lon hsdS_B_* (*r_B_*^−^*m_B_*^−^) λ(DE3 [*lacI* P*_lac_*_UV5_-*T7gene1 ind1 sam7 nin5*]) [*malB*^+^]_K-12_ (λ^S^)	[29]
CC118 λ*pir*+	Host strain for maintenance of *pir*-dependent recombinant plasmids	[30]
MG1655 Δ(ϕ80-*attB*)	MG1655 with deleted native (ϕ80-*attB*) site	[31]
DH5α	F-ϕ80*lacZ*∆M15 ∆(*lacZYA-argF*) U169 *recA1 endA1 hsdR*17 (r_k_^−^, m_k_+) *phoA supE*44 λ-thi-1 *gyrA*96 *relA*1	Laboratory collection
MG1655 Δ(ϕ80-*attB*) *IS*5.8::ϕ80-*attB*	MG1655 with deleted native ϕ80-*attB* site and reconstruction of *attB* site in *IS*5.8 locus	[31]
MG1655 *IS*5.8::P_L_-*hppa*^Rru^	MG1655 with deleted native ϕ80-*attB* site and *IS*5.8::P_L_-*hppa*^Rru^	This work
MG1655 *IS*5.8::P*_tac_*-*hppa*^Rru^	MG1655 with deleted native ϕ80-*attB* site and *IS*5.8::P*_tac_*-*hppa*^Rru^	This work
MG1655 *IS*5.8::P*_tac_*_21_-*hppa*^Rru^	MG1655 with deleted native ϕ80-*attB* site and *IS*5.8::P*_tac_*_21_-*hppa*^Rru^	This work
MG1655 ∆(ϕ80-*attB*) *adrA*::ϕ80-*attB*	MG1655 with deleted native ϕ80-*attB* site and artificial ϕ80-*attB* site in *adrA* locus	Laboratory collection
MG1655 ∆(ϕ80-*attB*) *adhE*::ϕ80-*attB*	MG1655 with deleted native ϕ80-*attB* site and artificial ϕ80-*attB* site in *adhE* locus	Laboratory collection
MG1655 *adrA*::P_L_-*hppa*^Rru^	MG1655 with deleted native ϕ80-*attB* site and *adrA*::P_L_-*hppa*^Rru^	This work
MG1655 *adrA*::P_L_-*hppa*^Rru^ ∆*ppa*::*cat*	MG1655 with deleted native ϕ80-*attB* site and *adrA*::P_L_-*hppa*^Rru^ and ∆*ppa*::λ*attR*-*cat*-λ*attL*	This work
MG1655 *adhE*::P_L_-*hppa*^Rru^	MG1655 with deleted native ϕ80-*attB* site and *adhE*::P_L_-*hppa*^Rru^	This work
MG1655 *adhE*::P_L_-*hppa*^Rru^ ∆*ppa::cat*	MG1655 with deleted native ϕ80-*attB* site, *adhE*::P_L_-*hppa*^Rru^ and ∆*ppa*::λ*attR*-*cat*-λ*attL*	This work
MG1655 *IS*5.8::P_L_-*hppa* ∆*ppa*::*cat*	MG1655 with deleted native ϕ80-*attB* site and *IS*5.8::P_L_-*hppa*^Rru^ and ∆*ppa*::λ*attR*-*cat*-λ*attL*	This work
MG1655 *IS*5.8::P_L_-*hppa* ∆*ppa*	MG1655 with deleted native ϕ80-*attB* site and *IS*5.8::P_L_-*hppa*^Rru^ and ∆*ppa*::λ*attB*	This work
MG1655 *IS*5.8::P_L_-*hppa*^Rru^ *adrA*::P_L_-*hppa*^Rru^	MG1655 with deleted native (ϕ80-*attB*) site, *IS*5.8::P_L_-*hppa*^Rru^ and *adrA*::P_L_-*hppa*^Rru^	This work
MG1655 *IS*5.8::P_L_-*hppa*^Rru^ *adrA*::P_L_-*hppa*^Rru^ ∆*ppa*	MG1655 with deleted native ϕ80-*attB* site and *IS*5.8::P_L_-*hppa*^Rru^, *adrA*::P_L_-*hppa*^Rru^ and ∆*ppa*::λ*attB*	This work
MG1655 *IS*5.8::P_L_-*hppa*^Rru^ *adrA*::P_L_-*hppa*^Rru^ *adhE*::P_L_-*hppa*^Rru^	MG1655 with deleted native ϕ80-*attB* site and *IS*5.8::P_L_-*hppa*^Rru^, *adrA*::P_L_-*hppa*^Rru^ and *adhE*::P_L_-*hppa*^Rru^	This work
MG1655 *IS*5.8::P_L_-*hppa*^Rru^ *adrA*::P_L_-*hppa*^Rru^ *adhE*::P_L_-*hppa*^Rru^ ∆*ppa*	MG1655 with deleted native ϕ80-*attB* site and *IS*5.8::P_L_-*hppa*^Rru^, *adrA*::P_L_-*hppa*^Rru^, *adhE*::P_L_-*hppa*^Rru^ and ∆*ppa*::λ*attB*	This work
pKD46	oriR101, repA101ts, *araC*, P*_araB_*-[γ, β, exo of phage λ], Ap^R^; used as a donor of λRed-genes to provide λRed-dependent recombination	[32]
pMWts-λInt/Xis	oriR101, repA101ts, λcIts857, λP_R_→λ*xis-int*, Ap^R^; used as a helper plasmid for thermoinducible expression of the λ *xis-int* genes	[31]
pAH123	oriR101, repA101ts, λcIts857, λP_R_→ϕ80-*int*, Ap^R^; used as a helper plasmid for thermoinducible expression of the ϕ80-*int* gene	[33]; GenBank accession number AY048726
pAH162-Tc^R^-2Ter	ϕ80-attP*,* pAH162, λ*attL*-*tetA*-*tetR*-λ*attR*	[31]; Gene Bank accession number AY048738
pAH162-Tc^R^-2Ter*-hppa*^Rru^	ϕ80-attP*,* pAH162, λ*attL*-*tetA*-*tetR*-λ*attR*, codon-harmonized [34] *hppa*^Rru^ from *R. rubrum*	This work
pUC57-*hppa*^Rru^	pUC57 low-copy plasmid, codon-harmonized *hppa*^Rru^ from *R. rubrum*	This work
pMW118-Cm^R^	oriR101, repA, MCS, Ap^R^, *λattR-cat-λattL*—donor of λXis/Int-excisable Cm^R^ marker	[35]
pMW118-Km^R^	oriR101, repA, MCS, Ap^R^, λ*attR*-*kan*-λ*attL*—donor of λXis/Int-excisable Km^R^ marker	[35]

^a^ VKPM, The Russian National Collection of Industrial Microorganisms.

**Table 2 microorganisms-11-00294-t002:** Sequences of the PCR primers used in this study.

Primer	Sequence 5′→3′	Description
P1	TGTAAAACGACGGCCAGT	Verification of the presence of chemically synthesized *hppa*^Rru^ gene by sequence analysis
P2	AGGAAACAGCTATGACCAT	Verification of the presence of chemically synthesized *hppa*^Rru^ gene by sequence analysis
P3	TCGAAGGAGGCAACGATTTCAGCTT	Amplification of the *ppa* gene for Southern hybridization
P4	TATTGAGATCCCGGCTAACGCAGAT	Amplification of the *ppa* gene for Southern hybridization
P5	CCTCCCTTTTCGATAGCGACAA	Verification of the presence of *hppa*^Rru^ gene in artificial ϕ80-*attB* site
P6	ACCGTTGGCGATCCGTACAA	Verification of the presence of *hppa*^Rru^ gene in artificial ϕ80-*attB* site
P7	TGGCCAGTGCCAAGCTTGCATGCCTGCAGCGCTCAAGTTAGTATAAAAAAGCTGAACGAGAAAC	Integration of phage lambda P_L_ promoter upstream *hppa*^Rru^ gene
P8	AGCGGCGGCTACGACGAAAAGATAGATGCCAGCCATAGTTAGTTCTCCTTCCGGCCAATGCTTCGTTTCG	Integration of phage lambda P_L_ promoter upstream *hppa*^Rru^ gene
P9	AACCGAAGCCCGGCGTTCAGGGTTATTACGCCAGAAGAACCGCTCAAGTTAGTATAAAAAAGCTGAAC	Amplification of the fragment for the *ppa* gene deletion
P10	CTCGGCACTTGTTTGCCACATATTTTTAAAGGAAACAGACTGAAGCCTGCTTTTTTATACTAAGTTGG	Amplification of the fragment for the *ppa* gene deletion
P11	TTACTAACCGAAGCCCGGC	Verification of the *ppa* gene deletion
P12	CGAAAACAAGCGAAGACATT	Verification of the *ppa* gene deletion

**Table 3 microorganisms-11-00294-t003:** Analysis of PPase activity in cytoplasmic fractions of *E. coli* strains expressing the *hppa*^Rru^ gene in the presence and absence of *E. coli ppa*.

Strain	PPase Activity, µmol min^−1^ mg^−1^
MG1655	5.6 ± 0.3
MG1655 *IS*5.8::P_L_-*hppa*^Rru^	5.5 ± 0.5
MG1655 *IS*5.8::P_L_-*hppa*^Rru^ ∆*ppa*	<0.5

Average data and standard deviations from 3 independent experiments are presented.

## Data Availability

The analyzed data presented in this study are included within this article. Further data are available upon reasonable request from the corresponding author.

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
