# Peer review of "H+-Translocating Membrane-Bound Pyrophosphatase from Rhodospirillum rubrum Fuels Escherichia coli Cells via an Alternative Pathway for Energy Generation"

_microorganisms, 2023, doi:10.3390/microorganisms11020294_

Round 1

Reviewer 1 Report

MAJOR

1. The main claim of the manuscript is that membrane-bound H(+)-PPase from Rhodospirillum rubrum can functionally replace the essential soluble PPase gene in E. coli. This finding would be novel and significant in the research field. However, the performed experiments did not go far enough to conclusively show that this indeed was the case. First, the authors did use only Northern blot to show that S-PPase gene positive band was eliminated in the deletion strain. More evidence could be shown, such as verification (or showing data if it was done) by PCR of the elimination of S-PPase in its native locus. Substantially stronger evidence, and these days not unreasonably costly (about $500/E.coli genome), would be to perform full genome sequencing of the mutant strain to show that S-PPase gene or (mutated) gene fragments are not present anywhere in the E. coli genome. The genomic analysis would also exclude the possibility of secondary mutations, which adapted the mutant E. coli to survive.
     Second, there is no evidence on protein level that the introduced H(+)-PPase gene was expressed. The accumulation of protein could be shown by Western analysis if the authors can get access to specific antibody. Alternatively, mass spectrometry could be used to show the presence of H(+)-PPase. Also RNA-level detection by RT-qPCR could provide further support and it is a sensitive technique to show also weak transcription.
     Third, there were attempts to show H(+)-PPase hydrolysis activity in isolated membrane vesicles (IMVs, Figure 5). The authors needed to use "ultrasensitive" fluorescence based Pi determination kit to observe any apparent PPi->2Pi activity and still the first indications of activity signal started to generate only after 20 min. Typically, H(+)-PPase activity in IMVs (when overexpressed from plasmid) can be detected without any difficulty by much less sensitive methods (papers from Lahti, Baykov and Goldman for example). Of course more enzyme expression is expected from plasmids but still the observed activity appears in significant in comparison to what one would expect for H(+)-PPase. Consistent with the reviewer's scepticism, Table 3 did not show any added PPase activity in S-PPase + H(+)-PPase strain or any detectable (insensitive assay, yes noted) H(+)-PPase activity in S-PPase negative strain; the sample should any way contain both soluble proteins as well as IMVs (low force centrifugation used did not pellet IMVs). Fluorescence signals in Figure 5 are also very prone to artefacts. The authors should convert the arbitrary units to meaningful enzyme activity units to allow quantitative assesment of H(+)-PPase activity. The papers from above mentioned labs indicate that plasmid-based expression of H(+)-PPases in E. coli typically lead to roughly 0.1-1 umol/min*mg PPase activity in IMVs. It is also alarming that the PPase activity in IMVs isolated from the strain with both S-PPase and H(+)-PPase genes did not statistically differ from the IMVs of wild-type E. coli. The explanation provided on this observation, e.g. competition for substrate by S-PPase (which was actually inhibited by fluoride), are not convincing. Indeed, as the authors perhaps also mean in their discussion of 'contamination', there is always residual S-PPase in IMV preparations, especially if they were not washed by multiple rounds of re-suspension/ultracentrifugation. Fluoride will not necessarily eliminate all this activity. When H(+)-PPase activity is weak it becomes difficult or even impossible to distinguish real H(+)-PPase activity from the background caused by S-PPase and other E. coli proteins (e.g. phosphatases/ATPases). Authors should provide the comprehensive analysis of inhibitor effects (fluoride for S-PPase, AMDP for H(+)-PPase) on measured PPase activities in IMVs, including statistical analysis, to exclude the alternative explanation that all apparent activities in Figure 5 are just random, unspecific background. If the authors cannot get hold of AMDP (specific inhibitor of membrane-bound PPases), it may be possible to kill H(+)-PPase by solubilizing IMVs with detergent (e.g. Triton X-100; H(+)-PPase typically does not survive IMV disruption without careful protective measures). In the current Figure 5 it is also unclear whether the mentioned three repeats are technical or three independent IMV preparations (as would be expected).

2. Results, line 538: "As expected, measurement of PPase activity in the strain with expression of heterologous H+-PPaseRru and deletion of the native ppa gene encoding S-PPase, confirmed the lack of the PPase activity in the soluble fraction (Table 3)."

This statement appears as overstatement as the sample contains both soluble proteins and IMVs. Furthermore, the detection limit of the assay seems so high that substantial PPase activity could go unnoticed.

3. Why OD595 and cell dry weight correlation, determined as part of metabolic flux analysis, is so different for E. coli with only H(+)-PPase gene? How does this change affect the calculations of growth properties and metabolic fluxes? Have authors tested the metabolic flux analysis in the scenario where the OD595 and cell dry weight correlation was the same for all strains, do the flux effects sustain. It appears as a reasonable request to perform further metabolic modeling by assuming OD595 and cell dry weight correlation being the same in all strains.

4. Related to #4, have authors observed the E. coli strains under microscope to find for further phenotypic changes? Would these observations explain the apparently changed OD595 and cell dry weight correlation?

5. Last sentence of Results: "The observed flux re-distribution confirmed that cells where native S-PPase was
replaced with H+-PPaseRru had a different metabolic state, which served as a basis for
testing the use of H+-PPaseRru for increasing the production of substances whose synthesis require high levels of energy."

It unclear what experiments authors mean by this. There are no such experiments done in this manuscript. If H(+)-PPase activity was hardly detectable (PPi hydrolysis) or not measured (proton pumping), how could the use of such H(+)-PPase strain promote the production of energy-demanding substances?  

6. Clarify what was the scar left into E. coli genomic DNA after the deletion of native, soluble PPase. Was S-PPase gene completely eliminated (all its gene sequence)? If some fragment of S-PPase gene was left into the genome, can it be excluded that it could encode partially functional PPase enzyme?

7. It seems that not all elements of the H-PPase gene expression unit are described, e.g., what was the transcription terminator at the end of inserted H-PPase gene. It would be beneficial to provide as Supplement the full annotated sequence of H-PPase expression unit, including also the regulatory elements (promoter, terminator) and other extra sequences incorporated in the mutant strain, where S-PPase was eliminated. A schematic / cartoon figure based description of the genomic structure of mutant strain(s) would also be beneficial in the main text.

8. Abstract. It goes a bit too far to speculate that the supposedly expressed H(+)-PPase promotes ATP synthesis by harvesting PPi free energy as proton gradient. H(+)-PPase activity was weak and proton pumping was not even shown.

MINOR

9. Indicate in the captions of Figures 3, 4 and 5 what are the error bars.

Intro: Membrane-bound PPases have their own EC numbers.
EC 3.6.1.1 - inorganic diphosphatase
7.1.3.1 H+-exporting diphosphatase
7.1.3.2 Na+-exporting diphosphatase

10. The legend of Figure 1. Scale bar mentioned in the caption is not shown. Correct typos.

11. Reference 15 is incomplete. Proof read all references for other mistakes.

12. Methods. To what buffer IMVs were resuspended after ultracentrifugation?

13. The legend of Figure 2 does not describe what is shown on the left and right gels. Would be better to aame them as A and B, and explain details in the legend.

14. There is variable use of hppa vs hppaRru, especially in figure legends. Make uniform throughout the manuscript.

Reviewer 2 Report

Malykh et al co-expressed the membrane-bound proton-pumping pyrophosphatase of R. rubrum  in E. coli. They found 36% reduced  TCA activity which is anticipated to be the cellular response on the improved ATP availability.

First, the reviewer wants to outline that he strongly supports the approach of the authors and he also appreciates their controls ensuring functionality of H+-PPaseRru in E. coli MG1655. However, the reviewer wonders about the unambiguousness of the results.

1) lines 540 - 542: What do the authors mean stating that 'same growth parameters' were achieved, outlining  - in the same sentence - that slower growth was observed? Actually, this growth phenotype is important for the entire discussion and should be presented in the main body of the text.

2) Figure 4 should consider MG655 hppa, too (see 1).

3) One of the key statements of the paper, namely the redistribution of carbon flux, is based on the analysis of MG1655-hppa-deltappa. (Please ensure the same strain naming in the entire paper including fig 1). Notably, this strain shows significantly reduced growth. Likely, biomass specific glucose uptake rates are reduced, too. Consequently, the question arises whether carbon flux patterns of said strain may be biased by the reduced growth rate preventing a comparison with the faster growing wild-type. Did the authors think of limiting carbon fluxes in the WT strain such that similar growth reduction occurs to ensure comparability of the results? This issue is not addressed so far and needs - at least - to be discussed for critically challenging the key conclusion.

3) The authors used laborious 13C labeling for identifying reduced TCA activity. Why didn't they analyse carbon dioxide emission rates? Actually, they should have this date otherwise the essential check for carbon balancing - a prerequisite for flux analysis - is missing.

4) The authors outline the benefits of heterologous pyrophosphate amplification. However, they do not show any measurement of e.g. ATP contents or ATP fluxes. The latter, might be derived from their flux map. In this context: Why did they assume maximum P/O ratios of 2 - a rather theoretical value which is acutually not used in the current analysis.

Round 2

Reviewer 1 Report

The clarifications in the text, small amount of added data (e.g. PCR-based verification of E. coli soluble PPase gene deletion) and metabolix flux analysis modelling under assumption of constant mass/optical density ratio in all strains have improved the manuscript significantly.

1. However, one major concern remains and it is whether reported PPase activity in Figure 5 is actually due to introduction H(+)-PPase of Rhodospirillum rubrum to E. coli. This was also one of the main concerns also in my original review report (point 1). The authors now clarified that activity measurements were actually only done for one vesicle (IMV) preparation per E. coli strain. For proving a biological hypothesis that H(+)-PPase can functionally replace native soluble PPase by providing sufficient PPi hydrolysis activity one would expect at least three biological replicates, i.e. three independent cell cultures per strain from which IMVs were isolated and activities determined, and accompanied statistical analysis (t-test or similar) to show that the activity profiles of the strains are different. It seems the experiment set would take only 1-2 weeks to complete.
     During IMV isolation protocol for the measurements suggested above the authors could also collect the first ultracentrifugation supernatant to obtain a sample that is the best for analyzing soluble PPase activity (if not too much diluted) because IMVs were quantitatively pelletted away. It would also be beneficial to determine soluble PPase activity in this soluble fraction and IMV fraction using the same high sensitivity fluorescent assay as in Figure 5 to provide clear cut comparison in which fraction (soluble/IMVs) the PPi hydrolysis activity of each strain resides. Especially as the reviewer disagrees with the Authors response #2 that relatively low force centrifugation ("The debris was removed by centrifugation at 12,000 rpm for 20 min (4 °C).") would remove IMVs from cell lysate. Almost all IMVs will remain in the supernatant and thus PPi hydrolysis activity in Table 3 is the sum of soluble PPase and H(+)-PPase activities.

2. Lane 631. "Interestingly, an unexplainably low rate of pyrophosphatase activity was detected in IMVs of the strain containing both types PPases, soluble and membrane-bound. At present, the reason underlying this effect is not clear, though could be related to  competitive interaction between M- and S-PPases toward substrates and limited access of the substrate to M-PPase in the presence of a soluble enzyme; moreover, the possibility of contamination in these experiments cannot be completely excluded."
     The explanation about substrate competition makes no sense as the PPase activity assay would detect both H(+)-PPase and soluble PPase activities as they both produce phosphate. Clarification or deletion of the speculation is needed. Are authors suggesting that the presence of soluble PPase in E. coli prevents expression of H(+)-PPase? This appears not to be case in typical overexpression experiments in which H(+)-PPase was expressed in E. coli from plasmid.

3. Table 3 and elsewhere. Specify what are error limits or error bars, e.g., standard deviation.

Reviewer 2 Report

The reviewer appreciates most of the answers given by the authors. However, the reviewer does not accept the general statement that 'strains, growing at different growth rates may be characterized by similar intracellular carbon distribution'.  The observed growth rates halved, Yxs changed, i.e. less precursors for biomass formation needed - and still the flux patterns are unchanged? Rather unlikely. Yxs clearly is a function of growth rate (and maintenance demands). If additional experiments are too laborious - which is accepted by the reviewer - authors should discuss the potential growth effect on the flux pattern, at least. 

Besides, the reviewer appreciates the consideration of additional growth curves. However, there seems to be a mismatch, now. Please check the number of strain labels given in the legend and the number of growth curves shown.
